# Clinical Profile of Patients with Idiopathic Pulmonary Fibrosis in Real Life

**DOI:** 10.3390/jcm12041669

**Published:** 2023-02-20

**Authors:** Diego Morena, Jesús Fernández, Carolina Campos, María Castillo, Guillermo López, María Benavent, José Luis Izquierdo

**Affiliations:** 1Servicio de Neumología, Hospital Universitario de Guadalajara, 19002 Guadalajara, Spain; 2Programa de Doctorado en Ciencias de la Salud, Universidad de Alcalá, 28801 Madrid, Spain; 3Diego Morena Valles, Servicio de Neumología Hospital Universitario de Guadalajara, Calle Donantes de Sangre, 19002 Guadalajara, Spain; 4SAVANA, 28013 Madrid, Spain; 5Departamento de Medicina y Especialidades Médicas, Universidad de Alcalá, 28801 Madrid, Spain

**Keywords:** IPF, artificial intelligence, real life

## Abstract

Objective: The objective of this study is to define the real-life clinical profile and therapeutic management of patients with idiopathic pulmonary fibrosis using artificial intelligence. Methods: We have conducted an observational, retrospective, non-interventional study using data from the Castilla-La Mancha Regional Healthcare Service (SESCAM) in Spain between January 2012 and December 2020. The Savana Manager 3.0 artificial intelligence platform was used to collect information from electronic medical records by applying natural language processing. Results: Our study includes 897 subjects whose diagnosis was compatible with idiopathic pulmonary fibrosis; 64.8% were men, with a mean age of 72.9 years (95% CI 71.9–73.8), and 35.2% were women, with a mean age of 76.8 years (95% CI 75.5–78). Patients who had a family history of IPF (98 patients; 12%) were younger and predominantly female (53.1%). Regarding treatment, 45% of patients received antifibrotic therapy. Patients who had undergone lung biopsy, chest CT, or bronchoscopy were younger than the patient population in whom these studies were not completed. Conclusions: This study has used artificial intelligence techniques to analyze a large population over a 9-year period and determine the situation of IPF in standard clinical practice by identifying the patient clinical profile, use of diagnostic tests and therapeutic management.

## 1. Introduction

Diffuse interstitial lung diseases (ILD) are a group of disorders with similar clinical, radiological and functional manifestations [1,2,3].

According to the 2013 ILD consensus statement by the American Thoracic Society/European Respiratory Society (ATS/ERS), the most frequent ILD is idiopathic interstitial pneumonia (IIP) [4,5,6], the most common of which is idiopathic pulmonary fibrosis (IPF) (40%) [7,8,9]. In recent years, the incidence of IPF has been on the rise [10,11], and a study carried out in the United Kingdom from 1968 to 2008 determined that the incidence of IPF is growing at a rate of 5% per year [12]. In Spain, the estimated number of patients with IPF is between 8000 and 12,000 [9,10].

It is important to know the family history of patients with IPF, as up to 20% of patients have a history of a family member with this pathology [4]. Other etiological factors that have been associated with the development of this disease are smoking, gastroesophageal reflux disease (GERD) associated with hiatal hernia in up to 80% and autoimmunity [1,7].

The symptoms and signs of patients with IPF are nonspecific, and a high clinical suspicion is therefore required for diagnosis [7]. The most frequent symptom is dyspnea, with insidious onset and variable evolution time [13,14]. IPF is usually diagnosed in patients over the age of 50, and the mean survival is 3 years after diagnosis [13,14].

The definitive diagnosis of IPF is made after other diffuse pulmonary parenchymal diseases have been ruled out [14,15]. Histologic or radiologic confirmation is also required. Diagnosis is confirmed by the presence of a histological pattern of usual interstitial pneumonia (UIP) observed in a surgical lung biopsy [15,16]. On imaging studies, evidence of a UIP pattern in high-resolution computed tomography (HRCT) with honeycomb or reticular abnormalities in both lung bases or the existence of traction bronchiectasis with decreased lung volumes are also considered high levels of diagnostic evidence for IPF [17,18,19]. The presence of pulmonary emphysema or pulmonary hypertension is a poor prognostic factor [20,21].

Over the last decade, the clinical management of patients with IPF has changed drastically due to the demonstrated damage that occurs with the so-called “triple therapy” (combination of prednisone, azathioprine and N-acetylcysteine) and with the arrival of the new era of antifibrotic therapies [22,23]. Two drugs (nintedanib and pirfenidone) were the first to be approved for the treatment of IPF, based on their efficacy in slowing the progression of the disease and reducing the annual risk of death [24,25,26]. These treatments may present adverse events or interactions with other drugs. Recent studies on nintedanib, a treatment that inhibits vascular growth factors (enhancing vascular dysfunction), have described a probable increased risk of bleeding in some patients [27,28,29].

Given all of the above, the main objective of this study is to determine the real-life clinical profile and therapeutic management of patients with idiopathic pulmonary fibrosis using artificial intelligence analysis with the Savana Manager 3.0 platform.

## 2. Materials and Methods

We designed an observational, retrospective and non-interventional study carried out in Castilla-La Mancha, Spain between 1 January 2012 and 31 December 2020 (Appendix A). Data were collected from the electronic health records (EHR) of patients diagnosed with IPF, and we followed the Strengthening the Reporting of Observational Studies in Epidemiology (STROBE) guidelines. The study was approved by the Research Ethics Committee of the Guadalajara public healthcare administration.

The total study population was 3,286,413 subjects, who had generated a total of 276,200,601 documents. For the analysis of patients with IPF, our study included all patients who met the inclusion criteria described in Table 1.

The data were extracted from the EHR thanks to a natural language processor using EHRead technology based on artificial intelligence and big data techniques—Savana Manager 3.0. This linguistic engine is capable of identifying unstructured clinical information (natural language or free text) from EHR, and subsequently transforming and formatting it into usable information for research purposes. The methodology used has been described previously [30,31]. Patient anonymity was maintained at all times. The data previously processed by the IT services of each hospital are anonymized and subsequently analyzed using computational linguistic techniques (SNOMED CT) [32]. Previously, the performance of SAVANA was evaluated for the correct identification of the IPF term. This evaluation of the system is calculated in terms of the standard metrics for precision (P), recall (R), and its harmonic mean F-score [33]. We obtained P, R and F-Scores for IPF of 1.0, 0.862 and 0.926, respectively. These results indicate that, within our population with IPF, all cases were accurately identified.

For the statistical analysis, the OpenEpi (v 3.0) and SPSS (v 25.0) applications have been used. Qualitative variables are expressed as absolute frequencies and percentages, while quantitative variables are expressed as means, 95%CI and standard deviations. For the analysis of the numerical variables, the independent-samples Student’s *t*-test was used. We applied the chi-square test to measure the association and compare proportions between qualitative variables. In all cases, differences whose *p*-value was less than 0.05 compared with the contrast test were considered significant.

This study has complied with the legal requirements described in the International Conference on Harmonization Good Clinical Practice guidelines. The guidelines of the Declaration of Helsinki (latest edition), Guidelines for Good Pharmacoepidemiology Practice and the data protection code for studies with Big Data and local regulations were followed. Informed consent was not required because the study is anonymous, observational and retrospective.

## 3. Results

Among the total study population, 897 subjects had been diagnosed with IPF. Figure 1 shows the flowchart of patients with IPF in this study, with a mean age of 74.2 (95%CI 73.2–75.1). In total, 581 (64.8%) were men with a mean age of 72.9 years (95%CI 71.9–73.8), and 316 (35.2%) were women with a mean age of 76.8 years (95%CI 75.5–78). A family history of IPF was reported in 98 patients (10.9%), 52 (53.1%) of whom were women, with a mean age of 50.8 years (95%CI 50–51.6).

Figure 2 shows the annual prevalence of IPF during the study period.

The symptoms described by these patients were mainly dyspnea (90.5%) and an ineffective cough (48.3%). Meanwhile, 231 (25.8%) patients of the study population had a smoking habit. The main comorbidities in our population are shown in Table 2. All the comorbidities described were significantly more prevalent in the population with IPF compared with the general population over 50 years of age.

The most commonly used diagnostic techniques in these patients were chest X-ray (92.8%) and chest HRCT (85.1%). Table 3 shows how simple bronchoscopy (33.8%) and surgical lung biopsy (14.1%), together with thoracic HRCT, were carried out in younger patients, whereas patients who did not undergo these diagnostic techniques were older.

Out of the total number of patients with IPF, 405 (45.2%) received antifibrotic treatment during the study period: 240 subjects were treated with pirfenidone (69.8 years; 95%CI: 68.6–70.9; 75.4% male) and 165 with nintedanib (70.4 years; 95%CI: 69.2–71.7; 72.7% male). The annual use of antifibrotic treatments throughout the study period is shown in Figure 3.

The population that was not treated with antifibrotic drugs (492 patients) had a mean age of 78.5 years (95%CI: 77.9–81.1). Table 4 shows an analysis between the untreated and treated populations and comorbidities. We observed a higher incidence of all comorbidities in the untreated population, except obesity, pulmonary emphysema and sleep apnea syndrome.

Table 5 presents the most frequent adverse events of antifibrotic treatment, highlighting gastrointestinal symptoms.

We specifically analyzed the risk of bleeding. Among patients treated with pirfenidone, ten (4.2%) experienced some type of bleeding. In addition, among the general population of patients treated with pirfenidone, eleven subjects (4.6%) were administered concomitant treatment with an oral anticoagulant drug (vitamin K antagonist or direct-acting anticoagulants), three (1.3%) of whom experienced bleeding. In these three patients, the time elapsed between the start of the drug therapy and the bleeding complication was 7, 62 and 12 days. In the population subgroup treated with nintedanib, five patients (3.0%) presented some type of bleeding (OR 0.72; 95%CI 0.22–2.13; *p* = 0.57). Among the patients treated with nintedanib, a total of three (1.8%) subjects received some type of concomitant oral anticoagulant treatment, only one of which developed a bleeding complication 362 days after starting treatment.

Lastly, 86 patients (9.6%) were candidates for lung transplantation.

## 4. Discussion

This study has used artificial intelligence (NLP) to analyze a large population over a period of 9 years to determine the situation of IPF in daily clinical practice. The results confirm the prevalence of this pathology in our setting, identifying a predominance in men over 70 years of age. The mean age of our population was higher in women. Compared with an annual increase of 5% in the United Kingdom between 1968 and 2008, our study demonstrated an average annual increase of 20%, although recent years have shown a trend towards stabilization [22]. This increase may have been due to the greater interest of doctors in this disease and not to an actual increased incidence of the disease in our setting, especially as a result of the appearance of effective antifibrotic drugs and the consequent creation of multidisciplinary interstitial disease committees in many hospitals [34].

Among the different IPF types, there is a specific group that has a familial form. The percentage of this family form ranges from 4% to 20% [4]. The age of onset is younger—usually before the age of 60. In our population, this percentage was slightly lower (10.9%) and more frequent in the female sex. The age of onset was also lower, with a mean age of around 50 years.

Moreover, non-familiar IPF is unusual below 50 years. By choosing this cut-off point, we have tried to reflect the wide range of the standard IFP population.

During the last decade, several etiological causes have been described that are potentially related to the appearance of IPF, although these have only been identified in 35% of cases [1]. It is also important to know which comorbidities these patients present, since many of them can function as etiopathological factors or increase complications. In our study, we have observed that patients with IPF present multiple comorbidities, as shown in Table 2. Regarding GERD or the presence of hiatal hernia, there is insufficient scientific evidence to confirm whether they are pathologically related to the development, progression or exacerbation of the disease over time; nonetheless, these conditions are present in a significant number of patients [1,7,15]. In our study, almost a quarter of the study population presented at least one of these pathologies.

In our setting, one out of four patients with IPF have a smoking habit. Xaubet et al [35]. described that this association occurs especially in smokers of more than 40 pack-years, with a mean age at presentation of 65 years. In this study, between 15% and 40% of patients who underwent lobectomy had pulmonary fibrosis. Although the role that tobacco plays in the pathogenesis of IPF is not well established, the prevention and treatment of smoking is essential to not increase the associated risks presented by these patients; even so, this is often not enough to prevent the progression of the disease [15].

The presence of pulmonary hypertension and emphysema drastically increases mortality in these patients. Smoking increases the risk of developing emphysema, mainly in patients with a high pack-year rate [35]. The prevalence of pulmonary fibrosis associated with emphysema is unknown, although it is estimated that between 5% and 10% of cases of diffuse interstitial lung disease are affected [21]. Pulmonary hypertension is a frequent complication in the clinical course of the association between emphysema and idiopathic pulmonary fibrosis, and it is the main condition that influences evolution and prognosis [20]. Previous studies have reported a prevalence of this association ranging between 47% and 90% [36]. In most published series, the diagnosis of pulmonary hypertension has been established by transthoracic echocardiography, which is a fundamental diagnostic technique to establish the evolution of the patient. In our study population, the prevalence of pulmonary hypertension and emphysema was much lower than in other national and international publications [20,21]. Being a population study, it is possible that the lower precision of the diagnostic process in older patients and in those who have not been considered for antifibrotic treatment could justify these differences. However, this prevalence is maintained in patients receiving antifibrotic treatment (Table 4).

We have observed a significant number of patients who have sleep apnea syndrome and associated IPF. The frequency of sleep-disordered breathing (SDB) in patients with pulmonary fibrosis remains controversial. Recent studies [37] have shown a high incidence, which even reaches 88% in certain series. In our study, the rate does not exceed 10%, possibly due to differences in population characteristics. However, it is important to rule out this pathology since it has been proven that nocturnal desaturations are an independent factor for mortality [15,37].

We have found no differences with the symptoms described in the bibliography; dyspnea and cough are the most frequent and present in more than 50% of patients [1,2].

The diagnosis is reached based on the findings of a radiological test, usually chest HRCT. In selected cases, histological confirmation by lung biopsy is required [1,2]. In our cohort, we observed that patients who underwent radiological or histological studies were younger (3–10 years, depending on the test) than patients who did not undergo extended testing.

The evolution of the treatment in these patients is represented in Figure 3, which demonstrates that, since the approval of pirfenidone in 2011 and nintedanib in 2015 (previously accepted as an orphan drug in 2014) by the European Medicines Agency, their use has increased progressively [24,25]. Since the use of these antifibrotic drugs and based on recent studies [27], fears have grown regarding the probable increase in adverse effects, especially the risk of bleeding associated with nintedanib by inhibiting vascular endothelial growth factors. In our population, this risk was not increased with the use of nintedanib, even with the concomitant use of antithrombotic drugs. Our population also presented other adverse effects associated with antifibrotic drugs, such as gastrointestinal symptoms. These results are in line with previous reports [25,26].

Data collection using NLP techniques has been possible thanks to the fact that all the clinical information of the Castilla-La Mancha health system has been digitized since 2012. However, as our study uses data from medical records, its main limitation is the lack of information recorded in the patient records. This lack of information may be more prevalent in the older population or those with a greater number of comorbidities, especially when antifibrotic treatment has not been started. For this reason, although some variables may not be correctly documented, this study only includes those in which the quality of the information has been verified. The analysis of this study was carried out with aggregated data, so the distribution of the sample cannot be known. Although the sample sizes of our study could make it unnecessary to check for normality, an approximation was made based on the confidence interval. Thus, the Z-score was calculated for each limit of the confidence interval. The Z-scores are within the limits of a standard normal distribution (−1.96 and 1.96 for a 95% confidence level).

## 5. Conclusions

This study identifies the main characteristics of an unselected IPF population, their comorbidities, the diagnostic tests performed, and the use of different antifibrotic therapies. Using artificial intelligence techniques (NLP), we are able to determine the situation of IPF in real life, without the possible selection biases that frequently occur in registry-based studies.

## Figures and Tables

**Figure 1 jcm-12-01669-f001:**
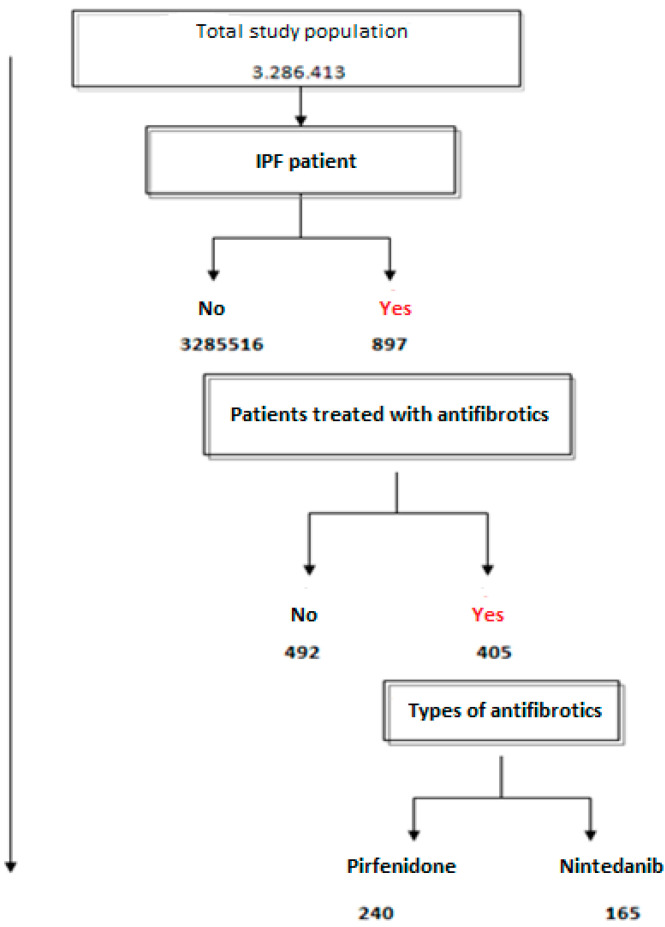
Flowchart showing the total number of patients in the study population, those with IPF and those with antifibrotic treatment.

**Figure 2 jcm-12-01669-f002:**
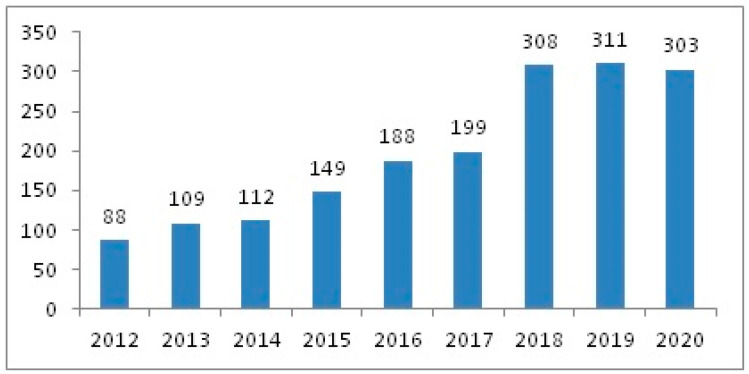
Annual prevalence of patients with IPF during the study period.

**Figure 3 jcm-12-01669-f003:**
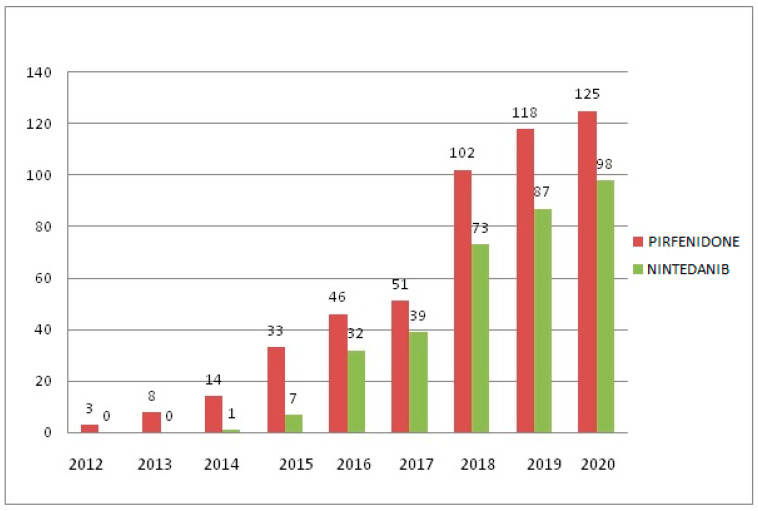
Annual evolution of antifibrotic treatment during the study period.

**Table 1 jcm-12-01669-t001:** Study inclusion and exclusion criteria.

Inclusion criteria
Patients with clinical diagnosis of idiopathic pulmonary fibrosis. The selected concept also includes idiopathic pulmonary fibrosis (acute fatal form), idiopathic pulmonary fibrosis, acute interstitial pneumonia and Hamman Rich syndrome
**Exclusion criteria**
Patients with a specific diagnosis other than idiopathic pulmonary fibrosis, including sarcoidosis, hypersensitivity pneumonitis, pulmonary edema, pneumonia, pulmonary embolism, pneumothorax, rib fractures, aspiration, pleural effusion or any other associated respiratory or non-respiratory disorder

**Table 2 jcm-12-01669-t002:** Comorbidities of patients with IPF and their prevalence versus the general population over the age of 50.

	IPF Patients	%	General Population	%	*p* Value/OR (95%CI)
N	897		1,410,880		
Mean (SD) age, years	74.2 (15.3)		68.3 (12.2)		<0.001
Male sex	581	64.8	657,470	46.6	2.1 (1.8–2.4)
Arterial hypertension	634	70.7	511,527	36.3	4.2 (3.7–4.9)
Dyslipidemia	504	56.2	350,926	24.9	3.9 (3.4–4.4)
Diabetes mellitus	363	40.5	241,004	17.1	3.3 (2.9–3.8)
Congestive heart failure	384	42.8	68,566	4.9	14.7 (12.8–16.7)
Hiatal hernia	201	22.4	81,452	5.8	4.7 (4.0–5.5)
Obesity	182	20.3	130,460	9.6	2.5 (2.1–2.9)
Pulmonary hypertension	250	27.9	34,014	2.4	15.6 (13.5–18.1)
Atrial fibrillation	209	23.3	81,207	5.8	4.9 (4.3–5.8)
Ischemic cardiopathy	200	22.3	47,669	3.4	8.2 (7.1–9.6)
Gastroesophageal reflux disease	112	12.5	46,153	3.3	4.2 (3.5–5.1)
Pulmonary emphysema	144	16.1	14,776	1.1	10.8 (8.7–13.4)
Sleep apnea syndrome	123	13.7	38,722	2.7	5.6 (4.7–6.8)

**Table 3 jcm-12-01669-t003:** Mean percentages of sex and age of patients who underwent chest CT, simple bronchoscopy or surgical pulmonary biopsy.

		SEX (Males, %)	OR(95%CI)	Mean Age (Years, 95%CI)	*p* Value
Chest CT	Yes (763)	66.2%	1.5 (1.1–2.1)	73.8 (95%CI 73–74.5)	*p* < 0.001
	No (134)	57.5%		76.2 (95%CI 73.7–78.7)	
Simple bronchoscopy	Yes (304)	64.8%	1.1 (0.8–1.4)	70.1 (95%CI 68.8–71.3)	*p* < 0.001
	No (593)	63.8%		75.8 (95%CI 74.9–76.7)	
Surgical pulmonary biopsy	Yes (127)	66.9%	1.1 (0.8–1.7)	66 (95%CI 64.3–67.8)	*p* < 0.001
	No (770)	64.2%		75.3 (95%CI 74.5–76)	

**Table 4 jcm-12-01669-t004:** Analysis of the main comorbidities between patients who were treated/not treated with antifibrotic therapy.

Comorbidities	Patients Treated	%	Patients Not Treated	%	OR (95%CI)
Arterial hypertension	235	58.0	399	81.1	0.3 (0.2–0.4)
Dyslipidemia	215	53.1	289	58.7	0.8 (0.6–1.0)
Diabetes mellitus	143	35.3	220	44.7	0.7 (0.5–0.9)
Obesity	84	20.7	98	19.9	1.1 (0.8–1.5)
Pulmonary hypertension	107	26.4	143	29.1	0.6 (0.4–0.8)
Heart failure	109	26.9	275	55.9	0.3 (0.2–0.4)
Mitral insufficiency	77	19.0	135	27.4	0.7 (0.5–0.9)
Ischemic cardiopathy	78	19.3	122	24.8	0.7 (0.5–0.9)
Emphysema	78	19.6	66	13.4	0.9 (0.6–1.2)
Atrial fibrillation	49	12.1	160	32.5	0.3 (0.2–0.4)
Sleep apnea syndrome	63	15.6	60	12.2	1.3 (0.9–1.9)

**Table 5 jcm-12-01669-t005:** Most frequent adverse events associated with the use of antifibrotics: nintedanib and pirfenidone.

Adverse Events	Nintedanib	%	OR (95%CI)	Pirfenidone	%	OR (95%CI)
Diarrhea	48	29.1	5.6 (3.6–8.7)	46	19.2	2.2 (4.5–3.3)
Nausea	20	12.1	2.8 (1.5–4.8)	25	10.4	1.1 (0.7–1.8)
Loss of appetite	28	17.0	5.1 (2.9–8.9)	27	11.3	1.2 (0.7–1.9)
Dizziness	37	22.4	5.2 (3.1–8.4)	48	20.0	2.2 (1.5–3.2)
Headache	15	9.1	2.8 (1.5–5.5)	21	8.8	1.6 (0.9–2.7)

## Data Availability

The data presented in this study are available upon request from the corresponding author. The data is not publicly available because it belongs to the national health system of Castilla La Mancha.

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
