# Peer review of "Clinical Profile of Patients with Idiopathic Pulmonary Fibrosis in Real Life"

_jcm, 2023, doi:10.3390/jcm12041669_

Round 1
Reviewer 1 Report
Dear Colleagues,
Thank you for your work. Unfortunately, in my opinion, major revision strongly required due to significant gaps and low quality of data presentation. Please find my comments below:
1. Please translate ALL figures (including drug names on Fig 3). Could you please also optimize data representation (e.g. 3 pairs of integer presented on Figure 1 should be provided in table form, Moreover you need only 4 numbers: Screened/ IPF no antifibrotic/IPF with pirfenidone/IPF with nintedanib)
2. Please completely redo the statistical processing of data.
a)Provide baseline characteristics for all population. Please compare baseline for IPF and general population (age, sex, smoking, desease duration etc)
b) Please review statistical criteria used evaluation each particular parameter. To use Student's t-test please check normality using appropriate criteria!
c) Please provide p-value for all cases. It will be nice if you provide Sensitivity analysis (you can use women's subpopulation if no other options can be used)
d) Please do the rounding in the same way for all cases (e.g. instead "98 patients (12%), 53.1% of whom were women" should be "98 patients (10.9%), 52 (53.1%) of whom 108 were women"
There are other problems, that can be discussed after major revision
Thank you.
Author Response
03 February 2023
Manuscript ID: jcm-2153700
Type of manuscript: Article
Title: CLINICAL PROFILE OF PATIENTS WITH IDIOPATHIC PULMONARY FIBROSIS IN
REAL LIFE
Authors: Diego Morena*, Jesús Fernández, Carolina Campos, María Castillo,
Guillermo López, María Benavent, José Luis Izquierdo
Comments to the Author from Section Manage Editor: Winnie Wu.
Thanks very much for the opportunity to re-submit a revised version of our manuscript to JCM. Please, find below a point-by-point response to all comments raised by the reviewers, to whom we are very grateful. We think that by introducing the requested changes, the ms. has improved. We hope that it may now be hopefully publishable in the JCM but we are open to continue working on it if needed. Thanks again.
Response to Reviewer 1 Comments
Thank you for your work. Unfortunately, in my opinion, major revision strongly required due to significant gaps and low quality of data presentation. Please find my comments below:
- We thank Rev. 1 for her/his assessment of our work. Below, we answer all the raised concerns individually. Thanks for these comments.
- Please translate ALL figures (including drug names on Fig 3). Could you please also optimize data representation (e.g. 3 pairs of integer presented on Figure 1 should be provided in table form, Moreover you need only 4 numbers: Screened/ IPF no antifibrotic/IPF with pirfenidone/IPF with nintedanib)
- Figures translation have been done
We respectfully disagree, so we believe that a flowchart is the standard for this type of data presentation. In fact, from the total population only those older than 50 years have been included for analysis, since IPF is very unusual in younger people. This data has been included.
- Please completely redo the statistical processing of data.
- a) Provide baseline characteristics for all population. Please compare baseline for IPF and general population (age, sex, smoking, disease duration etc)
- We included sex and age in table 2. Student t test has been performed for age. Due to huge number of patients ( 1 410 880) no previous test for normal distribution has been performed.
Some information such as severity or duration of treatment was not collected in this study. The reason was that only those variables well documented in the clinical files were selected.
- b) Please review statistical criteria used evaluation each particular parameter. To use Student's t-test please check normality using appropriate criteria!
- The analysis of this study has been carried out with aggregated data, so the distribution of the sample cannot be known. However, according to our statistical advisors, with the sample sizes that we are handling, it is not necessary to check normality. However, an approximation was made based on the confidence interval. For this, the Z-score was calculated for each limit of the confidence interval. The Z-scores are within the limits of a standard normal distribution (-1.96 and 1.96 for a 95% confidence level).
This comment has been included in limitations of the study. “The analysis of this study has been carried out with aggregated data, so the distribution of the sample cannot be known. Although the sample sizes that we are handling, could make unnecessary to check for normality, an approximation was made based on the confidence interval. For this, the Z-score was calculated for each limit of the confidence interval. The Z-scores are within the limits of a standard normal distribution (-1.96 and 1.96 for a 95% confidence level).
Figure 3 has been changed.
- c) Please provide p-value for all cases. It will be nice if you provide Sensitivity analysis (you can use women's subpopulation if no other options can be used)
- We use p values only when no IC95% are presented. For other analysis see above.
- d) Please do the rounding in the same way for all cases (e.g. instead "98 patients (12%), 53.1% of whom were women" should be "98 patients (9%), 52(53.1%) of whom 108 were women"
- Thank you for your comment. Changes have been done
There are other problems, that can be discussed after major revision
Thank you.
Thank you.

Reviewer 2 Report
In this study, Morena et al. assessed the clinical profile and therapeutic management of patients with IPF using artificial intelligence. I have the following comments:
- Figure 1 should be in English
- Have the authors an explanation for the relevant increase in IPF prevalence between 2017 and 2018? Was there a new drug approval for IPF in Spain?
- Why did the authors compare the IPF patients to a general population aged >= 50 and not with a comparable age of around 70 years?
- Where patients included once in the study or what was the mean duration?
- Did the authors get more information about IPF? Eg. age of onset, severity, duration of treatment.
Author Response
03 February 2023
Manuscript ID: jcm-2153700
Type of manuscript: Article
Title: CLINICAL PROFILE OF PATIENTS WITH IDIOPATHIC PULMONARY FIBROSIS IN
REAL LIFE
Authors: Diego Morena*, Jesús Fernández, Carolina Campos, María Castillo,
Guillermo López, María Benavent, José Luis Izquierdo
Comments to the Author from Section Manage Editor: Winnie Wu.
Thanks very much for the opportunity to re-submit a revised version of our manuscript to JCM. Please, find below a point-by-point response to all comments raised by the reviewers, to whom we are very grateful. We think that by introducing the requested changes, the ms. has improved. We hope that it may now be hopefully publishable in the JCM but we are open to continue working on it if needed. Thanks again.
REVIEWER 2
In this study, Morena et al. assessed the clinical profile and therapeutic management of patients with IPF using artificial intelligence. I have the following comments:
- We thank Rev. 2 for her/his assessment of our work and insightful comments, which we address individually below.
- Figure 1 should be in English
- Figures translation have been done
- Have the authors an explanation for the relevant increase in IPF prevalence between
- No treatment for IPF was approved in Spain in that period. Pirfenidone was introduced in our country in 2011 and nintedanib in 2015.
However, since its commercialization, a large amount of information has been produced that has been presented in different scientific environments and that has possibly been the cause of the greater alertness of doctors due to the availability of effective treatments.
- Why did the authors compare the IPF patients to a general population aged >= 50 and not with a comparable age of around 70 years?
- Although our population has a mean age of over 70 years, in the bibliography, previous studies have shown a mean age above 50 years. Also, an important subpopulation in our study are patients with a family history of IPF. These patients had a mean age of 50 years.
Moreover, non-familiar IPF is unusual below 50 years. By choosing this cut off point we try to reflect the wide range of the standard IFP population.
- Where patients included once in the study or what was the mean duration?
- The analysis was done by patients regardless the number of visits. We have not accurate information about the mean duration of the process.
- Did the authors get more information about IPF? Eg. age of onset, severity, duration of treatment.
- No other information such as severity, age at onset, or duration of treatment was collected in this study. The reason was that only those variables well documented in the clinical histories were selected.

Reviewer 3 Report
Very well written study with interesting outcomes. I think that the information presented will be of interest to others in the field and the AI data mining technique could be applied to other disease entities. I would request that the flow chart in Figure 1 be translated into English as the rest of the paper is written in this language.
Author Response
03 February 2023
Manuscript ID: jcm-2153700
Type of manuscript: Article
Title: CLINICAL PROFILE OF PATIENTS WITH IDIOPATHIC PULMONARY FIBROSIS IN
REAL LIFE
Authors: Diego Morena*, Jesús Fernández, Carolina Campos, María Castillo,
Guillermo López, María Benavent, José Luis Izquierdo
Comments to the Author from Section Manage Editor: Winnie Wu.
Thanks very much for the opportunity to re-submit a revised version of our manuscript to JCM. Please, find below a point-by-point response to all comments raised by the reviewers, to whom we are very grateful. We think that by introducing the requested changes, the ms. has improved. We hope that it may now be hopefully publishable in the JCM but we are open to continue working on it if needed. Thanks again.
REVIEWER 3
- Very well written study with interesting outcomes. I think that the information presented will be of interest to others in the field and the AI data mining technique could be applied to other disease entities. I would request that the flow chart in Figure 1 be translated into English as the rest of the paper is written in this language.
- We thank Rev. 3 for her/his assessment of our work
Translation done

Round 2
Reviewer 2 Report
My comments were addressed appropriately.
Author Response
Good night,
a review has been done by a professional translator who has introduced minor changes in the paragraphs that were included in response to the reviewers and in the rest of the article.
Thank you very much,
greetings.